# Inferring Excitatory and Inhibitory Connections in Neuronal Networks

**DOI:** 10.3390/e23091185

**Published:** 2021-09-08

**Authors:** Silvia Ghirga, Letizia Chiodo, Riccardo Marrocchio, Javier G. Orlandi, Alessandro Loppini

**Affiliations:** 1Center for Life Nano- & Neuro-Science, Istituto Italiano di Tecnologia (IIT), Viale Regina Elena 291, 00161 Roma, Italy; silvia.ghirga@iit.it; 2Engineering Department, Campus Bio-Medico University of Rome, Via Álvaro del Portillo 21, 00154 Roma, Italy; l.chiodo@unicampus.it; 3Institute of Sound and Vibration Research, Highfield Campus, University of Southampton, Southampton SO17 1BJ, UK; r.marrocchio@soton.ac.uk; 4RIKEN Center for Brain Science, Wako-shi 351-0198, Japan; javier.orlandigomez@riken.jp

**Keywords:** transfer entropy, local transfer entropy, synapses, calcium imaging, spiking, bursting

## Abstract

The comprehension of neuronal network functioning, from most basic mechanisms of signal transmission to complex patterns of memory and decision making, is at the basis of the modern research in experimental and computational neurophysiology. While mechanistic knowledge of neurons and synapses structure increased, the study of functional and effective networks is more complex, involving emergent phenomena, nonlinear responses, collective waves, correlation and causal interactions. Refined data analysis may help in inferring functional/effective interactions and connectivity from neuronal activity. The Transfer Entropy (TE) technique is, among other things, well suited to predict structural interactions between neurons, and to infer both effective and structural connectivity in small- and large-scale networks. To efficiently disentangle the excitatory and inhibitory neural activities, in the article we present a revised version of TE, split in two contributions and characterized by a suited delay time. The method is tested on in silico small neuronal networks, built to simulate the calcium activity as measured via calcium imaging in two-dimensional neuronal cultures. The inhibitory connections are well characterized, still preserving a high accuracy for excitatory connections prediction. The method could be applied to study effective and structural interactions in systems of excitable cells, both in physiological and in pathological conditions.

## 1. Introduction

Complex emergent dynamics in excitatory and inhibitory neuronal networks are at the basis of brain functions both in humans and lower-level organisms. In this context, increasing efforts have been devoted in recent years to the deep investigation of structure–function coupling, trying to link neuronal connectome to neurons synchronization and global network activity [1,2,3,4]. Indeed, structural connections naturally drive emergent dynamics that may result in different functional patterns depending on specific operating conditions, external stimuli, noise, and nonlinearities. On this basis, neuronal assemblies at different scales are investigated both in terms of structural networks and in terms of functional and effective networks, describing correlations and causal interactions among cells, both in in silico and in experimental studies. From an experimental point of view, comprehensive investigations of structure–function in neuronal assemblies have been possible thanks to advances in calcium imaging techniques able to precisely record activity from thousands of cells, simultaneously, despite suffering from limited temporal resolution with respect to multi-electrode array approaches [5,6,7,8,9,10].

Although functional and effective networks encode a different kind of information with respect to the structural connectome, data analysis approaches to infer functional/effective interactions can also be adopted to infer structural couplings under proper conditions. In the last years, the most used approaches to predict structural interaction were based on pairwise correlations between cells activity, partial correlations, Granger causality, and Transfer Entropy (TE) [11,12,13,14,15,16,17]. Among these techniques, TE, from information theory, has been shown to be a promising and accurate tool to predict structural interactions between neurons, and it was extensively applied to infer both effective and structural connectivity in small- and large-scale networks [15,18,19,20,21]. In the context of structural interactions inference, TE approaches generalized to account both for excitatory and inhibitory couplings have been developed and tested on in silico data, showing good performance in labelling structural connections [18,22]. However, the major drawbacks of those methods are the need for a double acquisition of network activity, successively blocking and non-blocking inhibitory interactions, the introduction of negative terms in the TE computation, and additional post-analysis based on TE features. In this contribution, we present an alternative method to infer excitatory and inhibitory structural interactions among neurons based on the splitting of the TE index into two contributions and a proper selection of the delay between the time series. We test our approach on simulated spike train data from small neuronal networks, investigating reconstruction performance for different network properties. Our results show that splitting of TE in local contributions and introducing a proper delay between source and target cells leads to accurate identification of inhibitory connections, preserving high accuracy of excitatory connection prediction in case of zero or minimal temporal delays. Our method can be potentially applied to infer both effective and structural interactions among neurons or in other biological systems, overcoming the limitations of the available methods. On this basis, our approach has potential impacts for a comprehensive investigation of organization principles of neural circuits both in physiological scenarios and in pathological conditions, including neurological diseases and disorders.

The paper is organized as follows: in Section 2, we describe our network model, including the setting of structural networks, the intrinsic cell dynamics, and briefly discussing the implementation details. Moreover, starting from the standard definition of TE, we introduce the TE splitting into local contributions encoding excitatory and inhibitory information flow. Furthermore, in Section 3, we present modeling results and the performance of local TEs in predicting excitatory and inhibitory synaptic connections. Finally, in Section 4 and Section 5, we discuss and comment on the predictive power of our generalized approach, also including open challenges to be explored in future contributions.

## 2. Materials and Methods

In this section, we present details of the modeled network in terms of structural features of neuronal assemblies and intrinsic cell dynamics. Then, by starting from the standard definition of TE as the main descriptor of the causal interactions, we introduce a local splitting of TE suited to dissecting excitatory and inhibitory information flow.

### 2.1. Neuronal Network Modeling

The overall network model comprises 100 neurons, in a 2D geometry, reflecting the structure of neuronal networks grown on plates for calcium imaging experiments. We consider these arrangements since most calcium imaging studies were performed on two-dimensional cultures due to the limited time resolution of 3D acquisitions. However, our approach is general and can be fully translated to volumetric cells assemblies. The position of neurons, assumed to be 0-dimensional, is randomly assigned on a square domain with sides of length 1, non-dimensional units. Outward connections from each cell are set through a spatial Gaussian kernel, with amplitude 1 on the reference cell and normally decaying to zero for increasing distances with respect to the reference cell, i.e., $K=\exp\left[-d_{ij}^{2}/\sigma_{r}^{2}\right]$, with *i* and *j* denoting the selected cells pair, and $d_{ij}$ the Euclidean distance between the cells. Specifically, synaptic projections are set by randomly extracting a number in the interval [0,1] and comparing it with the kernel amplitude at that specific distance. The standard deviation $\sigma_{r}$ of the kernel was set to 0.3 to achieve 20 inward connections per neuron on average, corresponding to a connection probability of 0.2. This setting was chosen to reproduce a spatial dependent connectivity similar to the one developed in cultures growing on plates and results in a mildly sparse network, in line with other modeling studies and data on cortical neuronal circuits [18,23].

The single neuron model responds to the need for an accurate description of electrical activity, coupled to a feasible computational cost for a network of hundreds or potentially thousands of neurons. The model [24,25,26] is a quadratic integrate-and-fire model with adaptation, able to reproduce the regular spiking and bursting behaviors of most cortical neurons. The differential equation describing the membrane voltage neuron dynamics is:(1)$$\frac{dv}{dt}=\frac{1}{\tau_{v}}\left[K_{v}\left(v-v_{r}\right)\left(v-v_{t}\right)-w+I_{S}\right]+g{}_{\xi }\xi \left(t\right)$$
with $\tau_{v}$ being the membrane voltage time constant; *w* the inhibitory current resuming the effect of slow current related to activation of potassium channels and inactivation of sodium channels; $I_{S}$ the synaptic current, coming as input from connected neurons; finally, ξ(t) denotes a white noise process with auto-correlation $\langle \xi \left(t\right)\xi \left(t^{\prime }\right)\rangle =\delta \left(t-t^{\prime }\right)$, and $g{}_{\xi }$ is the noise standard deviation. The quadratic part of Equation (Equation 1) includes the resting potential $v_{r}$ and the threshold potential $v_{t}$. If the stimulation is low, *v* remains lower that $v_{t}$, and the membrane potential relaxes towards its resting potential. After repeated stimulation, *v* can reach and exceed the threshold value, and can grow generating a spike ($v\geq v_{p}$). After the spike, the potential is reset to the value $v_{c}$ and the inhibitory current is reset to w+Δw.

The inhibitory current is described by the equation:(2)$$\frac{dw}{dt}=\frac{K_{w}\left(v-v_{r}\right)-w}{\tau_{w}}$$
where $\tau_{w}$ is the time characteristic of the inhibitory current *w*, and $K_{w}$ is the sensitivity of *w* to sub-threshold fluctuations of the membrane potential.

The membrane potential dynamics of each cell is monitored to check for spiking events, precisely recorded when *v* reaches the value $v_{p}$, just before resetting to control sub-threshold potential. Spike times $t_{s}$ are stored to compute the synaptic input on each neuron based on the pre-synaptic cells spiking history. Each contribution is delivered at time $t_{m}=t_{s}+d$, where *d* is a synaptic delay.
(3)$$\begin{array}{ccc} I_{S} & = & \sum_{j}\sum_{k}I_{jk} \end{array}$$
(4)$$\begin{array}{ccc} I_{jk} & = & g_{S}D_{j}\left(t_{sk}\right)\exp\left(1-\frac{t-t_{mk}}{\tau_{S}}\right)\left(\frac{t-t_{mk}}{\tau_{S}}\right)\theta \left(t-t_{mk}\right)\;. \end{array}$$

In Equations (Equation 3) and (Equation 4), *j* and *k* are indices scanning the whole sets of pre-synaptic cells and past spike times, respectively; $t_{sk}$ and $t_{mk}$ denote spike time and delayed spike time for the *k*-th spike of the *j*-th pre-synaptic cell; $g_{s}$ and $\tau_{S}$ represent strength and time constant of the synapse. The dynamical variable $D_{j}\left(t\right)$ describes depletion and renewal of neurotransmitter at the pre-synaptic cell. After each spiking event, *D* reduces to αD, with α∈[0,1], while, when no events occur, it is regulated by a dynamic tending exponentially to 1 to model the neurotransmitter replacement.
(5)$$\frac{dD(t)}{dt}=\frac{1-D}{\tau_{D}}\;.$$

We included AMPA and fast GABA-mediated currents (${\mathrm{GABA}}_{A}$), as those currents have a major role in regulating spiking and bursting dynamics in neuronal cultures. Our specific choice of the strengths for excitatory and inhibitory synapses is based on the amplitude of single post-synaptic evoked potentials (EPSP and IPSP), such as each pre-synaptic event which generates EPSP and IPSP of about 1.2 and 2 mV, respectively, [27,28,29]. We further amplified synaptic strengths by a factor ∼3–5 to account for the limited number of synaptic projections in our small networks [30]. By keeping the excitatory synaptic strength fixed, the inhibitory strength is varied in our simulations to simulate different balances of excitatory and inhibitory components. The reference balance of excitation and inhibition is $g_{E}/g_{I}$ equal to 1:2 (parameters reported below).

Other than intrinsic synaptic input, each neuron receives an incoming excitatory stimulus from an independent Poisson process with frequency λ, modeling the combined effect of extrinsic synaptic input and minis currents related to synaptic shot noise.

This random input serves as a basis to induce spontaneous activity in the network.

The model parameters used in numerical simulations are: $v_{r}=-60\;$mV, $v_{t}=-45\;$mV, $v_{p}=35\;$mV, $v_{c}=-50\;$mV, $\tau_{v}=50\;$ms, $K_{v}=0.5\;$mV ${}^{-1}$, $g_{\xi }=24.5\;$mV/ms, $\tau_{w}=50\;$ms, $K_{w}=0.5$, Δw=50mV, $g_{E}=200\;$mV, $g_{I}=400\;$mV, $\tau_{E}=1\;$ms, $\tau_{I}=5\;$ms, d=1 ms, α=0.8, $\tau_{D}=1000\;$ms, λ=0.5Hz. These values are fine-tuned to reproduce spiking/bursting behavior as observed in neuronal cultures.

The network model is resolved with an in-house C++ code, implementing a fourth-order Runge–Kutta scheme with a fixed time step of Δt=0.1ms, opportunely evolving both the deterministic and stochastic terms for the membrane potential dynamic. Total simulation time was set to 5 min.

Since our interest is to apply TE on signals closely resembling the ones acquired from calcium imaging techniques, we computed calcium signals from simulated the membrane potential time series by convolving spike trains as calculated from numerical simulations with a calcium response kernel, $A\;\exp\left(-t/\tau_{\mathrm{Ca},2}\right)\left(1-\exp\left(-t/\tau_{\mathrm{Ca},1}\right)\right)$, where A=1, $\tau_{\mathrm{Ca},2}=700\;$ms, $\tau_{\mathrm{Ca},1}=10\;$ms [18,26,31,32]. This function models an exponential rise and decay of calcium at every spike event, with different on and off time scales set to reproduce experimental signals. We also added a white noise process with a mean of 0 and a standard deviation of 10% with respect to *A* to introduce noise as usually observed in the optical acquisition of calcium data, despite not directly modeling calcium binding to fluorescence indicators. Further, both noisy and deterministic calcium traces were down-sampled at 5, 10, and 20 ms to analyze TE performance on coarse calcium traces as usually acquired in experiments (∼100 Hz frame rate). In line with network analysis performed on experimental calcium, we focused our investigation on binary spike train signals opportunely extracted from down-sampled calcium data. Specifically, the coarse calcium traces are taken as a starting point for a further spike detection whose result is the input to the TE analysis. Spike identification on calcium signals is performed through an accurate analysis of signal derivative, leading to an almost perfect reconstruction of the coarse spike train in case of deterministic down-sampled calcium signal, and to a ∼90% accurate identification of spikes in case of noisy coarse calcium. To acquire this accuracy, additional conditions are imposed related to the derivative at later times with respect to the reference time to discard random fluctuations [33]. Specifically, we identify an onset and an offset: at the onset, the derivative must exceed a certain positive threshold. At the offset, the derivative must be below a certain negative threshold. We labeled all time points between each detected pair of onset and offset with 1, considering neurons as active throughout the rising phase.

It is worth noting that also in the case of deterministic coarse calcium signals, the reconstructed spike train is not exactly equal to the down-sampled spike train as computed by numerical integration of the network, since recognition by derivative tends to extend by a small amount of time the activated state of the neuron respect to the spiking event.

### 2.2. Transfer Entropy

For two discrete Markov processes $X=[x_{1},x_{2},\ldots ,x_{T}]$ and $Y=[y_{1},y_{2},\ldots ,y_{T}]$ the TE from *Y* to *X* is defined as [34]:(6)$$TE_{Y\to X}=\sum_{x_{n},x_{n-1}^{\left(k\right)},y_{n-d}^{\left(k\right)}}P\left(x_{n},x_{n-1}^{\left(k\right)},y_{n-d}^{\left(k\right)}\right)\;\log\frac{P(x_{n}|x_{n-1}^{\left(k\right)},y_{n-d}^{\left(k\right)})}{P(x_{n}|x_{n-1}^{\left(k\right)})}$$
where $x_{n}$ is *n*-th element of *X*, $x_{n-1}^{\left(k\right)}$ and $y_{n-d}^{\left(k\right)}$ are the *k*-dimensional vectors $\left[x_{n-1},\ldots ,x_{n-k}\right]$ and $\left[y_{n-d},\ldots ,y_{n-d-k+1}\right]$, respectively. For simplicity, in Equation (Equation 6) *k* denotes Markov order both for *x* and *y*, but in the following $k_{x}\neq k_{y}$ will be assumed. TE quantifies the flow of information directed from *Y* to *X*. TE is defined on transition probabilities and represents the distance in the probability space between the single-node transition matrix $P(x_{n}|x_{n-1}^{\left(k\right)})$ and the two-nodes transition matrix $P(x_{n}|x_{n-1}^{\left(k\right)},y_{n-d}^{\left(k\right)})$ [35]. It ranges from 0 to infinity: non vanishing values indicate the existence of causality from *Y* to *X*, i.e. past values of *Y* are helpful in predicting the future behavior of *X*, while 0 values indicate lack of causality, i.e. single node and two nodes transition matrix are identical.

To infer effective neuronal network connectivity, TE and its extensions have been successfully applied to various types of neuronal data, in which *X* and *Y* are binary time series corresponding to activities of two neuronal units (the values are labeled by 1 when neurons are spiking and by 0 when at rest). The efficiency of TE in quantifying information transfers among neuronal units is proved by several studies: TE has been exploited to elucidate general physical principles of neuronal networks, such as the structure–function relationship [15,16,18,36], small-worldness and scale-free properties [37,38], existence of hubs and modules [39], etc.

However, TE defined as in Equation (Equation 6) cannot distinguish between inhibitory and excitatory synapses. For this purpose, the aim of this work is to combine a generalized version of TE [15], suitable for calcium imaging data, with a proper splitting in local excitatory and inhibitory contributions. This step is fulfilled by defining appropriate subsets of states related to excitation and inhibition, in line with a recent study focusing on the identification of inhibitory and excitatory connections from multi-electrode array data [22]. This distinction of connections is fundamental to understanding how a small number of inhibitory neurons functionally interplay with a large number of excitatory neurons controlling network dynamics, bursts and synchronization [40,41,42,43].

The key idea is to consider local TEs, which estimate the information transfer between events rather than variables [44], thus we split TE in Equation (Equation 6) into the two contributions, the excitatory one and the inhibitory one, defined, respectively, in Equations (Equation 7) and (Equation 8).
(7)$$TE_{E_{Y\to X}}=\sum_{\begin{array}{c} x_{n},x_{n-1}^{\left(k\right)},y_{n-d}^{\left(k\right)}:\\ x_{n}=y_{n-d}^{\left(k\right)} \end{array}}P\left(x_{n},x_{n-1}^{\left(k\right)},y_{n-d}^{\left(k\right)}\right)\;\log\frac{P(x_{n}|x_{n-1}^{\left(k\right)},y_{n-d}^{\left(k\right)})}{P(x_{n}|x_{n-1}^{\left(k\right)})}$$
(8)$$TE_{I_{Y\to X}}=\sum_{\begin{array}{c} x_{n},x_{n-1}^{\left(k\right)},y_{n-d}^{\left(k\right)}:\\ x_{n}\neq y_{n-d}^{\left(k\right)} \end{array}}P\left(x_{n},x_{n-1}^{\left(k\right)},y_{n-d}^{\left(k\right)}\right)\;\log\frac{P(x_{n}|x_{n-1}^{\left(k\right)},y_{n-d}^{\left(k\right)})}{P(x_{n}|x_{n-1}^{\left(k\right)})}$$
In the excitatory component, the sum goes over the states such that $x_{n}$ and $y_{n-d}^{\left(k\right)}$ have the same event; that is, they are both active or inactive. In the inhibitory component instead, the sum is restricted over the states such that $x_{n}$ and $y_{n-d}^{\left(k\right)}$ have the opposite events, i.e., one is active and the other inactive. It is important to note that, in the above definitions of excitatory and inhibitory subsets of states, $y_{n-d}^{\left(k\right)}$ is considered active (equal to 1) if at least one of its entries is 1. Instead, all the time series $y_{n-d}$,..., $y_{n-d-k+1}$ are included for the evaluation of the joint probability distribution function and local TEs.

In our splitting, particular care should be taken for combinations of states such as {$y_{n-d}^{\left(k\right)}=0$,$x_{n}=0$}, {$y_{n-d}^{\left(k\right)}=0$,$x_{n}=1$}. We assume that the first occurrence is linked to a hidden excitatory contribution since no activity in one neuron corresponds to no activity on the other neuron. Instead, the second occurrence underlies a hidden inhibitory connection since the silenced state of the inhibitory source is unable to stop the activity on the target cell. We could modify our assumptions and treat those contributions differently, but computed results show that these subsets promote a consistent identification of excitatory and inhibitory information. Furthermore, our assumptions on excitatory and inhibitory contributions are in line with those presented in the literature [22].

To apply this approach to calcium imaging data, taking into account the general characteristics of the system, TE is modified in two main aspects: the inclusion of “same-bin” interaction and the selection of dynamical states [15]. Since typical temporal resolutions available for calcium imaging recording (>5 ms) are greater than excitatory synaptic time constants (∼1 ms) and comparable with inhibitory ones (∼5 ms), excitatory connections are better identified if considering causal interactions between events that occur within the same time-bin, that is the case of d=0. Slower interactions can be still included by calculating TE for a Markov order greater than 1. Our approach aims to further investigate this aspect by evaluating excitatory and inhibitory contributions of TE for several time bin-sizes (5 ms, 10 ms, 20 ms) both in case of zero and non-zero delay, with fixed Markov orders $k_{x}=1$, $k_{y}=2$. We verify that while taking into account same-bin interaction improves the detection of an excitatory connection, the inhibitory component of TE is better highlighted by looking at delayed anti-correlation, which is mainly addressable to the longer timescale of inhibitory currents. Additionally, the restriction of TE evaluation to non-synchronous dynamical states is crucial to properly capture interactions between neurons. Selection of dynamical states is performed by analyzing the histogram of the average signal of the network and choosing a threshold value in order to include in the analysis all data points at time instants in which the average fluorescence is below the selected threshold. This procedure is equivalent with respect to the use of a conditioning variable to select regions of interests (ROIs) for the analyzed signals [15]. The presented results in the following section are related to the best accuracy obtained for different selections of ROIs, performed by varying the threshold between the minimum and maximum value of the average calcium trace. Despite a slight variability, all the selections of ROIs we tested gave accurate results, with the only exception of the more restrictive threshold close to the minimum value of the average calcium.

## 3. Results

We assess the performance of our algorithm on calcium fluorescence traces generated by simulating dynamics of a modeled neuronal network including 100 cells—80% excitatory and 20% inhibitory—connected via a spatial Gaussian kernel (Figure 1A, see Methods). We analyze data simulated from 10 realizations of the model with the same spatial distribution of cells but a different sorting of excitatory and inhibitory neurons. Network dynamics are simulated at a different balance between excitation and inhibition in terms of synaptic conductance (the ratio $g_{E}/g_{I}$). Emergent activity in a control condition, i.e., $g_{E}/g_{I}$ equal to 1:2, consists of spiking/bursting oscillations with frequency of ∼0.5–1 Hz, and presents both synchronous and asynchronous regimes (Figure 1B). Calcium traces, computed by a convolution of the spike train with a calcium response kernel, are down-sampled at 5, 10, and 20 ms bin size, enriched with noise, and further binarized with derivative-based methods (Figure 1C–E, see Methods). Spike trains extracted from calcium traces are used to calculate excitatory and inhibitory contributions of TE in Equations (Equation 7) and (Equation 8), for different values of the delay *d* and $k_{x}$ and $k_{y}$ fixed to 1 and 2, respectively, (Figure 1F). TE components are tested separately in predicting excitatory and inhibitory connections. Standard Receiver-Operator Characteristic (ROC) analysis is exploited to quantify the distance between inferred effective connectivity and structural ones by comparing TE matrices to the corresponding structural adjacency matrices. In particular excitatory (inhibitory) adjacency matrix has entry $a_{ij}$ equal to 1 if exist a structural excitatory (inhibitory) link from *j* to *i* and 0 otherwise. ROC curves are generated by considering a variable threshold from the lowest to the highest TE value, and taking as significant indices only the ones above the threshold. At each threshold, values are then computed and plotted the fractions of true positives (TPR) and false positives (FPR). In Figure 2A, Figure 3A and Figure 4A the mean and the standard deviation of the ROC curves obtained for the 10 realizations of the network are reported. A reliable detection is obtained for simultaneous low values of FPR and high values of TPR; thus the area under the curve (AUC) is an indicator of the accuracy: the greater it is the better is the detection. We also evaluate the Youden’s J, as the maximal value of TPR−FPR, and the corresponding sensitivity (TPR) and specificity (1-FPR), reported for all cases examined in Table 1, Table 2 and Table 3. In Figure 2B, Figure 3B and Figure 4B, we compare local excitatory and inhibitory components of TE in different operating conditions: the excitatory component of TE at 0 − bin delay is plotted versus the inhibitory component evaluated at different delays. Black lines denote the best thresholds for predictions whose evaluation is based on the Youden’s J in the independent ROC analysis in the panels A. Left and bottom density plots represent projections of the two-dimensional distributions of TE indices on excitatory and inhibitory components, respectively. Such a graph provides the intuitive idea of the ability of the local TE components to distinguish different types of connections.

### 3.1. Increasing the Delay Improves Inhibitory Network Reconstruction

Our main goal is to address the role of *d* on the accuracy of the network reconstruction, with particular regard to inhibitory connections. On this basis, we analyzed in detail local excitatory and inhibitory TE indices at different offsets of the source signal with respect to the target cell. Figure 2 displays results for 0, 1 and 2-bin delays (d=0, 1, 2) both for the excitatory and for inhibitory local components of TE (red and blue, respectively). The down-sampled bin size is 10 ms and the ratio $g_{E}/g_{I}$ is set to 1:2. The ROC curves in Figure 2A show that at increasing delays, the local excitatory component of TE decays in performance while the local inhibitory component improves. In particular, we note that for delay values greater than 0, in terms of bins, the inhibitory ROC (blue curves) displays a sharp rise at low FPR, indicating a significant increase of the accuracy with respect to the case of zero delay in which TPR is always near to the FPR (Table 1). In Figure 2B we compare the local excitatory component of TE at 0-bin delay with the inhibitory component evaluated for the different delays considered. Delays greater than 0 in the inhibitory part give rise to improved segregation and more accurate reconstruction of real spatial connections. Indeed, excitatory links, represented by red spots, are localized on the top left of the graph, exhibiting high excitatory TE and low inhibitory TE values, while inhibitory connections (blue spots) are localized on the right, being well sorted above the threshold of the inhibitory TE component.

To further investigate the effect of noise on spike recognition from calcium down-sampled data and consequent effects on TE performance, we repeat the same analysis with spike trains as extracted from noisy coarse calcium signals. Computed results in terms of ROC parameters are shown in Table 1 and show a non-significant loss in detection accuracy, still leading to an efficient reconstruction of both excitatory and inhibitory connections optimized at the same values of *d* as observed in the deterministic signals analysis.

### 3.2. The Ratio $g_{E}/g_{I}$ Does Not Affect Network Reconstruction

Similar results are obtained varying the ratio $g_{E}/g_{I}$, as shown in Figure 3 for ratio equal to 1:1, 1:2, 1:3 and same down-sampled bin size set to 10ms. Note that from here on, the three delays overlap in the same figure. ROC curves in Figure 3A exhibit high values of TPR corresponding to low values of FPR, both for excitatory (with d=0, 1) and for inhibitory (with d=2) component, regardless of the ratio $g_{E}/g_{I}$. An increasing imbalance towards inhibition slightly improves accuracy, sensitivity, and specificity of inhibitory synapses recognition, but still, control condition and unbalance towards excitation leads to accurate and good results (Table 2). Figure 3B also confirms a good partition of excitatory and inhibitory links analogous to the one observed in Figure 2B for the inhibitory component calculated with a delay d=2.

### 3.3. Down-Sampled Bin Size Effect on Network Reconstruction

Figure 4 depicts results varying the down-sampled bin size between the values of 5, 10, 20ms and for a ratio $g_{E}/g_{I}$ fixed to 1:2. In the case of the smaller bin size, 5ms, we observe for the local excitatory and inhibitory TE quite similar behavior with respect to the case of 10ms bins as previously analyzed. However, small differences can be noticed (see Figure 4A and Table 3): excitatory TE is less affected by delay variation since bin size gets smaller; both components preserve good accuracy, but surprisingly, the best performance is obtained with a bin size of 10ms. This means that increasing temporal resolution in our approach is not only not necessary for efficient structural link detection, it actually makes it worse. An improvement could be achieved by increasing the Markov order $k_{y}$, but this implies higher computational costs. On the contrary, for 20ms bins, the ROC curves show a more drastic performance decay of the local excitatory component of TE at increasing delays. Moreover, prediction of inhibitory links results in higher accuracy for 1-bin delays and worse for 2-bin delays. Figure 4B shows that good segregation of excitatory and inhibitory connections can be achieved for each bin size by choosing an appropriate delay.

## 4. Discussion

Reconstruction of structural connectivity among neurons from experimental recordings of their activity is a key aspect to unveil organization principles of neuronal networks, both in simple and complex organisms, with the aim to shed light on the structure–function coupling in those complex biological systems. Despite several studies largely explored a variety of techniques in inferring functional, effective, and structural connectivity, mostly based on cross-correlations, causalities, and information theory measures [11,12,13,14,15,16,17], only a few recent works attempted to generalize these approaches to label excitatory and inhibitory synapses selectively [17,18,22]. Specifically, these methods try not only to infer the network connectome but also the sign of each connection. However, among the ones based on TE, which is currently assumed to be one of the best indices to infer connections and causal dependencies, also accounting for nonlinear interactions, the presented techniques show some drawbacks, including the need for additional post-analysis or multiple recordings of neuronal activity, with and without specific chemicals to suppress inhibition. In this contribution, we propose an alternative method to infer excitatory and inhibitory links among neurons trying to overcome the abovementioned limitations. Our approach consists of a proper splitting of TE in local contributions, encoding excitatory and inhibitory information transfer. We define the subsets of states related to the excitatory and inhibitory combination based on the evidence that inhibitory links promote a “silenced” state in the target while excitatory links an “active” state. This choice is also in line with similar approaches proposed in the literature [22]. Computed results on in silico networks show that considering a proper delay between the source and the target improves the accuracy of detection substantially for inhibitory connections, reaching levels of accuracy comparable to the current approaches based on TE but avoiding similar drawbacks. In our model, the optimal delay is in the range of 10–20 ms and is in line with the time scales of fast GABA-mediated inhibitory currents [45,46,47,48,49]. Furthermore, the detection of excitatory links is more accurate at zero-bin delays, denoting higher excitatory information transfer at physical time delays in the range of 1-10 ms, which is consistent with the fast activation of AMPA-activated currents [50,51,52]. It is worth mentioning that our analysis of 0, 1, and 2-bin delays is mainly due to the down-sampled calcium data used as starting point. The main reason for this is because neuronal activity in in vitro and in vivo experiments is mostly recorded via calcium imaging techniques that allow for efficient simultaneous acquisition of neuronal activity from thousands of cells, with a time resolution usually limited at 1 ms in the best situations. However, based on the very general formulation of our local TE measures, our approach is potentially applicable to other kinds of data, such as multi-electrode array recordings, by opportunely redefining delays and Markov orders in the TE computation. A further improvement here presented with respect to similar applications of TE in dissecting structural connections concerns the mathematical model adopted to reproduce neurons dynamics. Indeed, previous applications of TE on synthetic calcium data were mostly based on simple integrate-and-fire models with limited dynamical behavior compared to the quadratic integrate-and-fire, which also accounts for bursting response other than spiking activity.

Of course, in future implementations, the adopted model could also be replaced by more detailed biophysical descriptions of single neuronal dynamics [53,54] also accounting for intracellular calcium balance to avoid the use of a predefined calcium kernel to simulate network response. Furthermore, our investigation of excitatory and inhibitory information transfer should also be tested in light of different models of synaptic dynamics, possibly accounting also for NMDA- and slow GABA-activated synaptic currents [55,56]. In this regard, preliminary investigations of different formulations of post-synaptic currents, based on double exponentials suited to set specifically activation and deactivation time constants, fine-tuned to reproduce experimental data, show that our modeling and TE results are qualitatively unchanged, still showing a significant improvement in the detection of inhibitory synapses when a proper delay is taken into account in the local TE measures. On this basis, we believe that our approach can be a valuable alternative for the complete dissection of neuronal connectome both in small and large-scale systems, in in silico and real experiments. Particular care must be considered in the case of application to real experimental data. Indeed, the best accuracy of local TE in our simulations is achieved by comparison with the ground truth network structure, a piece of information that is usually not available in experiments and represents the outcome to be predicted most of the time. In this case, surrogate data should be considered to test for the statistical significance of the computed TE indices [57,58,59], eventually filtering only the most significant ones to identify actual information transfer. Given our approach and techniques for significance testing, extension to real data is immediate, and near future studies will be devoted to the selection of optimal surrogate models to identify actual communications between neurons. Other than the prediction of excitatory and inhibitory structural connectivities, we believe that our approach is potentially applicable also to infer effective connectivity among neurons. In this context, it is well known that a unique network structure in neuronal assemblies gives rise to various dynamical regimes, depending on external inputs, intrinsic parameters, or noise. Strongly synchronized and asynchronous activities are among the type of emergent activities that can arise in neuronal networks. By starting from recorded activity, TE reconstructed connectivity is sensible to dynamical regimes of the network, as explored in previous studies [15], and some dynamical states, such as the asynchronous regime, are favorable in predicting structural connections while are less accurate in case of synchronous emergent activity. In this scenario, we believe that local TE can still be a useful indicator of functional and effective connectivity despite switching the goal from the inference of structural links to the identification of causal interactions as observed from dynamics. On this basis, the approach proposed here has the potential to be applicable to the inference of structural and effective connectivity in in vitro and in vivo neuronal circuits with a strong impact on the comprehension of basic mechanisms underlying information acquisition, storage, and transmission in physiological or pathological scenarios. Particular relevance in this regard concerns the investigation of network properties in neurological disorders or pathologies, such as schizophrenia, Alzheimer’s disease, strokes and epilepsy [60,61,62,63,64], where our approach can be useful not only in predicting general features of connectivity but also in identifying the unbalance between excitation and inhibition.

## 5. Conclusions

In this work, we formulate an alternative method to infer structural excitatory and inhibitory connections between neurons and test its accuracy on a network model representative of real neuronal assemblies. The basic idea is to dissect TE in local excitatory and inhibitory components and analyze their relative contribution at different communication delays. Our results indicate that via TE splitting and considering specific delays, it is possible to discriminate excitatory and inhibitory synaptic couplings. Our approach has the potential to be generalized to other kinds of recordings of neuronal activity and can also be applied to the inference of effective connectivity. Future investigations should be devoted to validating our approach against experimental data and other biophysical models of neuronal network activity. Future computational studies should also deal with comparative benchmarks, including other available methods to selectively identify excitatory and inhibitory connections starting from the recorded activity.

## Figures and Tables

**Figure 1 entropy-23-01185-f001:**
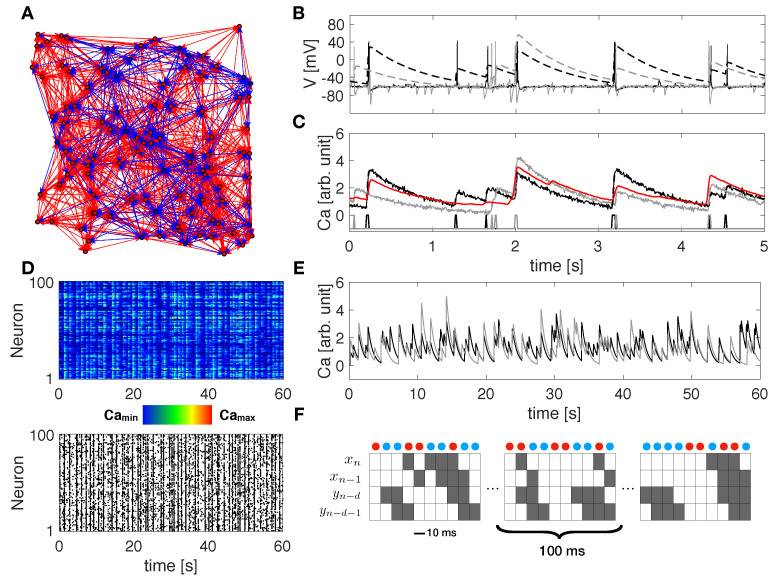
Modeled networks, simulated data and TE decomposition.(**A**) Example of modeled neuronal network including 100 cells, 80% excitatory and 20% inhibitory, connected via a spatial gaussian kernel. Red/blue circles and arrows denote excitatory/inhibitory neurons and directed connections, respectively. (**B**) Simulated membrane voltage time series (continuous curves) and corresponding calcium signals (dashed curves) for two representative cells (black and gray). Calcium traces are computed by a convolution of the computed spike train with a calcium response kernel. (**C**) Calcium traces (black and gray) after down-sampling at 10 ms bin size and noise addition. At the bottom, spike trains as extracted from the down-sampled calcium traces. The red curve shows the average calcium response of the network. (**D**) Raster plot showing down-sampled calcium traces (top) and spike events (bottom) over the entire network in 1 min, as detected from the down-sampled calcium traces. (**E**) 1-min down-sampled calcium traces (black and gray) for the same representative cells shown in panels (**B**,**C**), as computed from the convolution with the original spike train. (**F**) TE decomposition scheme on two representative spike trains. Red and blue dots denote combinations of state encoding excitatory and inhibitory information from *Y* to *X*, respectively. The parameter *d* represents the delay considered on the source *Y*, in terms of bins.

**Figure 2 entropy-23-01185-f002:**
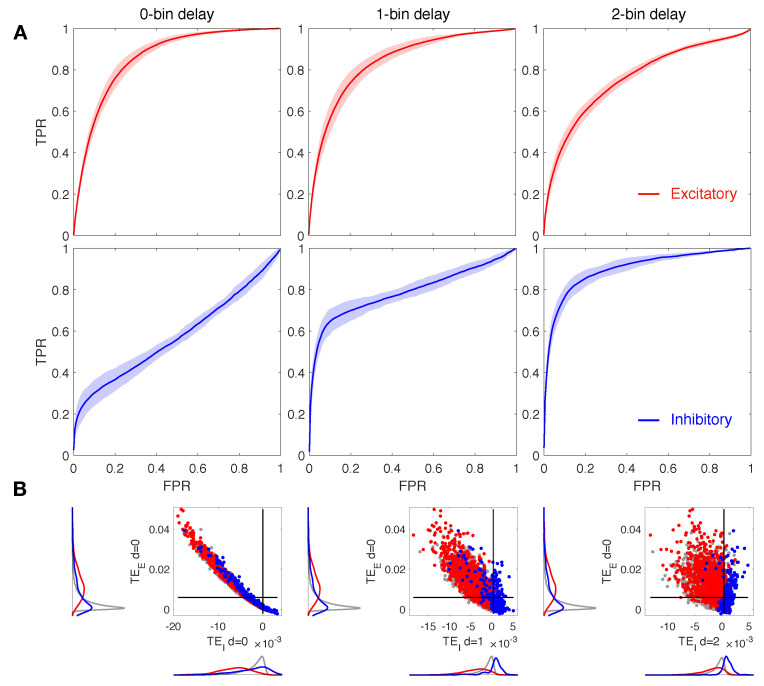
Reconstruction of structural connections at varying delays. (**A**) ROC curves showing the accuracy of excitatory and inhibitory components of TE (red and blue, respectively) in predicting real synaptic connections. ROC analysis is conducted over 10 realizations of the model, based on the same spatial distribution of cells but a different sorting of excitatory and inhibitory neurons. Continuous red and blue lines and corresponding light red and blue areas denote mean and standard deviation, respectively, computed over all the simulations. TE is evaluated on spike trains extracted from the calcium traces down-sampled at 10 ms. TE components are tested separately in predicting excitatory and inhibitory connections by comparing TE matrices to the corresponding structural adjacency matrices encoding solely excitatory or inhibitory links. The local excitatory component of TE decays in performance at increasing delays, while the local inhibitory component increases in performance at increasing delays. (**B**) Comparison of local excitatory and inhibitory components of TE in a single modeled network. The excitatory component of TE at 0-bin delay is plotted versus the inhibitory component evaluated at different delays. Delays greater than 0 in the inhibitory part give rise to improved segregation and more accurate reconstruction of real spatial connections. Gray dots denote TE indices for uncoupled cells, while red and blue dots denote TE indices for excitatory and inhibitory real connections. Black lines denote the best thresholds for predictions as evaluated in the independent ROC analysis of panel (**A**), based on the Youden’s J. Left and bottom density plots represent projections of the two-dimensional distributions of TE indices on excitatory and inhibitory components, respectively.

**Figure 3 entropy-23-01185-f003:**
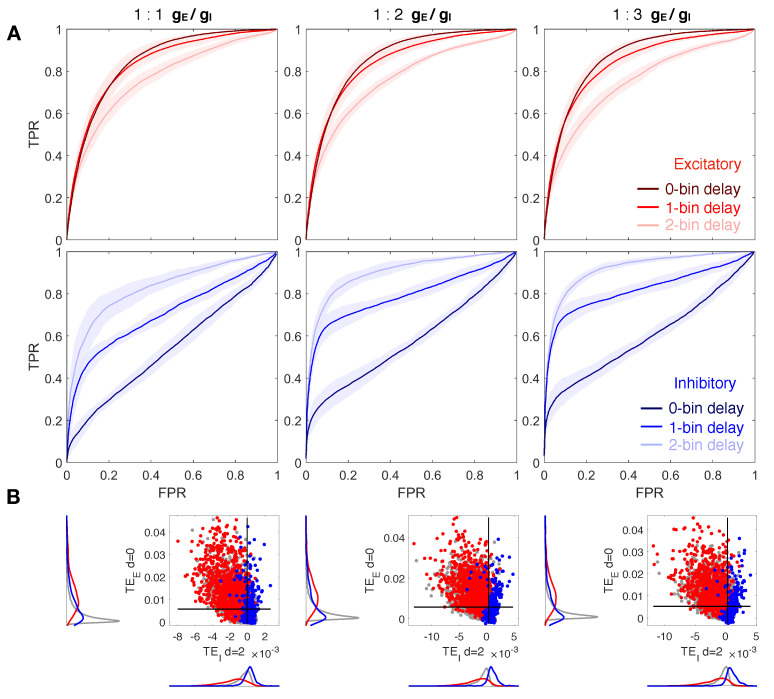
Reconstruction of structural connections at a varying ratio $g_{E}:g_{I}$, and at different delays. (**A**) ROC curves showing the accuracy of excitatory and inhibitory components of TE (red and blue, respectively) in predicting real synaptic connections. ROC analysis is conducted over 10 realizations of the model, based on the same spatial distribution of cells but a different sorting of excitatory and inhibitory neurons. Continuous red and blue lines and corresponding light red and blue areas denote mean and standard deviation, respectively, computed over all the simulations. Dark, medium and light red/blue curves represent ROC at different delays for excitatory/inhibitory connections. TE is evaluated on spike trains extracted from the calcium traces down-sampled at 10 ms. TE components are tested separately in predicting excitatory and inhibitory connections by comparing TE matrices to the corresponding structural adjacency matrices encoding solely excitatory or inhibitory links. Left, central, and right columns show results at a varying balance between excitation and inhibition in terms of synaptic conductance ($g_{E}$:$g_{I}$): 1:1 (left), 1:2 (central, control case), 1:3 (right). In every case, the local excitatory component of TE decays in performance at increasing delays, while the local inhibitory component increases in performance at increasing delays. (**B**) Comparison of local excitatory and inhibitory components of TE in a single modeled network. The excitatory component of TE at 0-bin delay is plotted versus the inhibitory component evaluated at 2-bin delay, giving the best segregation and more accurate reconstruction of real spatial connections. Gray dots denote TE indices for uncoupled cells, while red and blue dots denote TE indices for excitatory and inhibitory real connections. Black lines denote the best thresholds for predictions as evaluated in the independent ROC analysis of panel (**A**), based on the Youden’s J. Left and bottom density plots represent projections of the two-dimensional distributions of TE indices on excitatory and inhibitory components, respectively.

**Figure 4 entropy-23-01185-f004:**
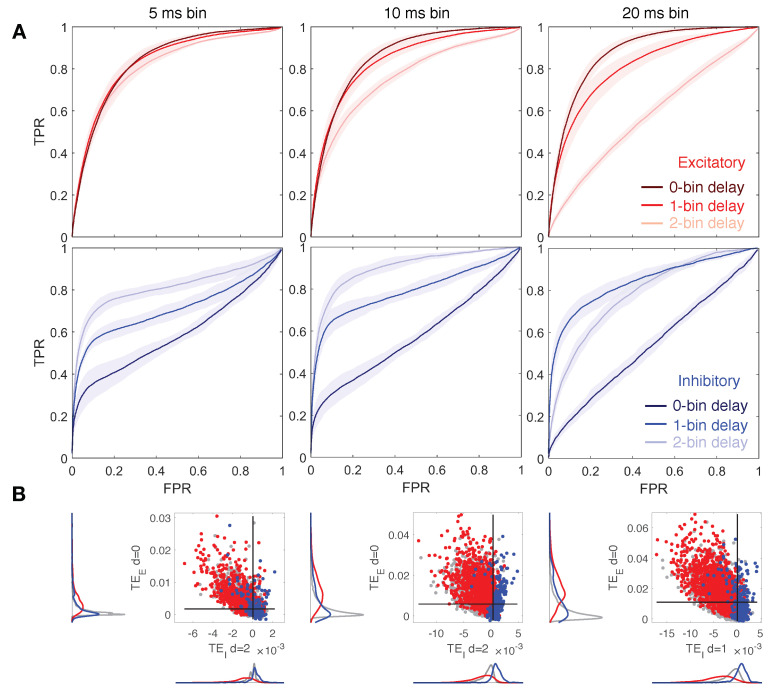
Reconstruction of structural connections at a varying down-sampled bin size, and at different delays. (**A**) ROC curves showing the accuracy of excitatory and inhibitory components of TE (red and blue, respectively) in predicting real synaptic connections. ROC analysis is conducted over 10 realizations of the model, based on the same spatial distribution of cells but a different sorting of excitatory and inhibitory neurons. Continuous red and blue lines and corresponding light red and blue areas denote mean and standard deviation, respectively, computed over all the simulations. Dark, medium and light red/blue curves represent ROC at different delays for excitatory/inhibitory connections. Networks dynamic is simulated at a ratio $g_{E}$:$g_{I}$ equal to 1:2. TE components are tested separately in predicting excitatory and inhibitory connections by comparing TE matrices to the corresponding structural adjacency matrices encoding solely excitatory or inhibitory links. Left, central, and right columns show results at a varying down-sampled bin size: 5 ms (left), 10 ms (central, control case), 20 ms (right). For the majority of cases, the local excitatory component of TE decays in performance at increasing delays, while the local inhibitory component increases in performance at increasing delays. At 20 ms bin size, 1-bin delay results in higher accuracy in predicting inhibitory links. (**B**) Comparison of local excitatory and inhibitory components of TE in a single modeled network. The excitatory component of TE at 0-bin delay is plotted versus the inhibitory component giving rise to the best segregation of real connections. Gray dots denote TE indices for uncoupled cells, while red and blue dots denote TE indices for excitatory and inhibitory real connections. Black lines denote the best thresholds for predictions as evaluated in the independent ROC analysis of panel (**A**), based on the Youden’s J. Left and bottom density plots represent projections of the two-dimensional distributions of TE indices on excitatory and inhibitory components, respectively.

**Table 1 entropy-23-01185-t001:** Effect of noise in the identification of synaptic connections. Average AUC, Youden’s J, and corresponding sensitivity and specificity over 10 neuronal networks (see Section 3). Reconstruction starting from signals down-sampled at 10 ms bin size. Balance between excitation and inhibition $g_{E}/g_{I}$ is set to 1:2. Both in deterministic and noisy data, ROC analysis shows equivalent performance in links prediction.

	Deterministic [bin 10 ms, $g_{E}/g_{I}$ 1:2]						Noise [bin 10 ms, $g_{E}/g_{I}$ 1:2]					
	E			I			E			I		
d	**0**	**1**	**2**	**0**	**1**	**2**	**0**	**1**	**2**	**0**	**1**	**2**
**AUC**	0.86	0.84	0.76	0.58	0.79	0.90	0.86	0.84	0.76	0.58	0.76	0.89
**J**	0.57	0.54	0.41	0.21	0.55	0.68	0.57	0.54	0.41	0.19	0.46	0.65
**Sens**	0.83	0.76	0.63	0.30	0.65	0.81	0.83	0.79	0.68	0.29	0.57	0.80
**Spec**	0.75	0.79	0.78	0.91	0.90	0.87	0.74	0.76	0.73	0.90	0.89	0.84

**Table 2 entropy-23-01185-t002:** Effect of excitation inhibition balance in the identification of synaptic connections. Average AUC, Youden’s J, and corresponding sensitivity and specificity over 10 neuronal networks. Reconstruction starting from signals down-sampled at 10ms bin size. Balance between excitation and inhibition $g_{E}/g_{I}$ is set to 1:1, and 1:3. Unbalance toward excitation leads to worse results in the identification of inhibitory links, however, still preserving good performance at 2-bin delays. Unbalance toward inhibition improve inhibitory links identification keeping unaltered identification accuracy for excitatory connections.

	Deterministic [bin 10 ms, $g_{E}/g_{I}$ 1:1]						Deterministic (bin 10 ms, $g_{E}/g_{I}$ 1:3]					
	E			I			E			I		
d	**0**	**1**	**2**	**0**	**1**	**2**	**0**	**1**	**2**	**0**	**1**	**2**
**AUC**	0.84	0.84	0.76	0.54	0.71	0.83	0.86	0.84	0.75	0.60	0.82	0.92
**J**	0.55	0.54	0.42	0.12	0.39	0.56	0.58	0.54	0.39	0.26	0.60	0.71
**Sens**	0.82	0.79	0.67	0.32	0.53	0.72	0.83	0.75	0.64	0.33	0.69	0.85
**Spec**	0.73	0.75	0.75	0.80	0.86	0.84	0.75	0.79	0.75	0.93	0.91	0.86

**Table 3 entropy-23-01185-t003:** Effect of varying bin size in the identification of synaptic connections. Average AUC, Youden’s J, and corresponding sensitivity and specificity over 10 neuronal networks. Reconstruction starting from signals down-sampled at 5 and 20ms bin size. Down-sampling at 5 ms has minimal effect on excitatory links prediction accuracy, slightly reducing performance in the detection of inhibitory connections. Down-sampling at 20 ms significantly deteriorates excitatory links prediction at increasing bin delays and highlights a loss performance for the identification of inhibitory synapses at 2-bin delays.

	Deterministic [bin 5 ms, $g_{E}/g_{I}$ 1:2]						Deterministic (bin 20 ms, $g_{E}/g_{I}$ 1:2)					
	E			I			E			I		
d	**0**	**1**	**2**	**0**	**1**	**2**	**0**	**1**	**2**	**0**	**1**	**2**
**AUC**	0.84	0.83	0.81	0.59	0.71	0.81	0.87	0.81	0.58	0.54	0.84	0.78
**J**	0.54	0.54	0.50	0.27	0.46	0.59	0.61	0.48	0.13	0.10	0.57	0.42
**Sens**	0.82	0.79	0.74	0.34	0.56	0.72	0.85	0.74	0.51	0.31	0.71	0.69
**Spec**	0.72	0.75	0.77	0.93	0.90	0.88	0.76	0.75	0.62	0.79	0.86	0.73

## Data Availability

Not applicable.

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
