# Peer review of "Inferring Excitatory and Inhibitory Connections in Neuronal Networks"

_entropy, 2021, doi:10.3390/e23091185_

Round 1

Reviewer 1 Report

Title: “Inferring excitatory and inhibitory connections in neuronal networks “

In this manuscript the authors presented a revised version of Transfer Entropy (TE) which was able to efficiently excitatory and inhibitory neural activities. This procedure was tested over in silico small neuronal networks, built to simulate the calcium activity as measured via calcium imaging in two-dimensional neuronal cultures. The authors claim that inhibitory connections are well characterized, still preserving a high accuracy for excitatory connections prediction. Finally, the authors claim that the method could be applied to study effective and structural interactions in systems of excitable cells, both in physiological and in pathological conditions.

General comment: This manuscript is well written and organized. A revised way to calculate TE is provided to disentangle excitatory and inhibitory neural activities in a 2D network. This seems to be a novel contribution to better understand and predict structural interactions among neurons. Even if the network is simulated in silico and the real applications of this procedure are a future issue to explore, this work seems to be an interesting contribution to the journal. Panels are good and also the language level seems to be adequate.

Some detailed comments:

Lines: “The paper is organized as follows. In Section 2, we describe our network model,

63 including the setting of structural networks, the intrinsic cell dynamics, and briefly dis-

64 cussing the implementation details. Further, starting from the standard definition of TE,

65 we introduce the TE splitting into local contributions encoding excitatory and inhibitory

66 information flow. In Section 3, we present modeling results and the performance of local

67 TEs in predicting excitatory and inhibitory synaptic connections. Finally, in Sections 4

68 and 5, we discuss and comment on the predictive power of our generalized approach,

69 also including open challenges to be explored in future contributions.”

*)Perhaps the text should read as: “ The paper is organized as follows: in Section 2, we……. Moreover, starting from the standard definition of TE,……. Furthermore, in Section 3, we present modeling results a…… Finally, ….. “

lines: “2.1. Neuronal Network modeling.

76 The overall network model comprises 100 neurons, in a 2D geometry, reflecting the

77 structure of neuronal networks grown on plats for calcium imaging experiments.”

*) The authors could better explain why a 2D geometry was chosen to reflect the structure of neural networks which are 3D. In other words, they should better explain why are relevant neural networks grown on 2D surfaces for calcium imaging experiment. And, more specifically, what are the maiin hypotheses trough which a network of 3D cells grown in a 2D surfaces could be modelled as a 2D network.

Lines: “The position of neurons, assumed 0-dimensional, is randomly assigned on a square

79 domain with side 1, non-dimensional units. Outward connections from each cell are set

80 through a spatial Gaussian kernel, with amplitude 1 on the reference cell and normally

81 decaying to zero for increasing distances with respect to the reference cell, i.e., K =

82 exph!d2ij/sr2i, with i and j denoting the selected cells pair, and dij the Euclidean distance

83 between the cells. Specifically, synaptic projections are set by randomly extracting a

84 number in the interval [0,1] and comparing it with the kernel amplitude at that specific

85 distance. The standard deviation sr of the kernel was set to 0.3 to achieve 20 inward

86 connections per neuron on average, corresponding to a connection probability of 0.2”

*) The authors should also explain why these mathematical details reflect the biological characteristics of a real network of neurons.

Lines: “The differential equation describing the membrane voltage

94 neuron dynamics is:…..…..…...

95 with tv being the membrane voltage time constant; w the inhibitory current resuming

96 the effect of slow current related to activation of potassium channels and inactivation

97 of sodium channels; IS the synaptic current, coming as input from connected neurons;

98 finally, x(t) denotes a white noise process with auto-correlation hx(t)x(t0)i = d(t ! t0).

99 The quadratic part of Eq. 1 includes the resting potential vr and the threshold potential

100 vt. If the stimulation is low, v remains lower that vt, and the membrane potential relaxes

101 towards its resting potential. After repeated stimulation, v can reach and exceed the

102 threshold value, and can grow generating a spike (v % vp). After the spike, the potential

103 is reset to the value vc and the inhibitory current is reset to w + Dw.”

*) Perhaps, the meaning of g_{\xi} is currently lacking in Eq. (1)…..

lines: “Model parameters used in numerical simulations are: vr = !60 mV, vt = !45 mV,v
133 p = 35 mV, vc = !50 mV, tv = 50 ms, Kv = 0.5 mV!1, gx = 24.5 mV/ms, tw = 50 ms,
134 Kw = 0.5, Dw = 50 mV, gE = 200 mV, gI = 400 mV, tE = 1 ms, tI = 5 ms, d = 1 ms,
135 a = 0.8, tD = 1000 ms, l = 0.5 Hz.
136 The network model is resolved with an in-house C++ code, implementing a fourth-
137 order Runge-Kutta scheme with a fixed time step of Dt = 0.1 ms, opportunely evolving
138 both the deterministic and stochastic terms for the membrane potential dynamic. Total
139 simulations time was set to 5 minutes.
140 Since our interest is to apply TE on signals closely resembling the ones acquired from
141 calcium imaging techniques, we computed calcium signals from simulated membrane
142 potential time series by convolving spike trains as calculated from numerical simulations
143 with a calcium response kernel, A exp(!t/tCa,2)(1 ! exp(!t/tCa,1)), where A = 1,
144 tCa,2 = 700 ms, tCa,1 = 10 ms [18,26,31]. “

*) The authors should better explain to the interested readers why just these values have been chosen and what are the biological reasons of this choice.

*) Could the authors better explain the main characteristics of the calcium response kernel ?

lines; “. Particular care must be considered in the case

390 of application to real experimental data. Indeed, the best accuracy of local TE in our

simulations is achieved by comparison with the ground truth network structure, a piece

392 of information that is usually not available in experiments and represents the outcome

393 to be predicted most of the time. In this case, surrogate data should be considered to test

394 for the statistical significance of the computed TE indices [56–58], eventually filtering

395 only the most significant ones to identify actual information transfer. Future studies

396 will be devoted to the application of our approach on such experimental data and on

397 the selection of optimal surrogate models to identify actual communications between

398 neurons. “

*) The authors should better explain these lines to the interested readers. How far is the application of this methods to real experimental data ?

Reviewer 2 Report

Dear Authors,
The article proposes an alternative method for inferring structural excitatory and inhibitory connections between neurons. Its accuracy was tested on a network model representative of real neuronal assemblies. The approach is new and very interesting. The article is inspiring, excellently written and composed, and it contains significant results and directions for further research. It is difficult to start a discussion with eminent specialists in this subject. The article is carefully written and completely prepared for publication in the journal. 

Some suggestions for consideration:
- a full stop at the end of the article title is an oversight;
- in the abstract (line 10): it sounds better to use "in the article we present ..." instead of "we present here ...";
- line 242: it should be "We assess" instead of "We asses" which is missing one letter "s";
- line 254: it should be "Spike train extracted from calcium traces is used to" instead of "Spike train extracted from calcium traces are used to" - the problem is that "train" (singular) is used with the plural form of the verb "to be" or else it should be "trains";

In the final conclusion, I do not think that any substantive corrections should be made.

Round 2

Reviewer 1 Report

The authors revised the manuscript according to the comments of this reviewer.